# Proceedings from the First Onco Summit: LATAM Chapter, 19–20 May 2023, Rio de Janeiro, Brazil

**DOI:** 10.3390/cancers16173063

**Published:** 2024-09-03

**Authors:** Vania Hungria, Anna Sureda, Garcia Rosario Campelo, Marco Aurélio Salvino, Karthik Ramasamy

**Affiliations:** 1Hematology, Faculty of Medical Sciences of Santa Casa de São Paulo, São Paulo 01224-001, Brazil; hungria@dialdata.com.br; 2Clinical Hematology Department, Catalan Institut Català d’Oncologia-L’Hospitalet, Instituto de Investigación Biomédica de Bellvitge (IDIBELL), University of Barcelona (UB), 08908 Barcelona, Spain; asureda@iconcologia.net; 3Thoracic Tumors Unit, Medical Oncology Department, University Hospital A Coruña Biomedical Research Institute (INIBIC), 15006 A Coruña, Spain; ma.rosario.garcia.campelo@sergas.es; 4Cell Therapy, D’OR Institute Research & Education (IDOR)/PPGMS-Federal University of Bahia (UFBA), Salvador 40110-100, Brazil; marcohemato@hotmail.com; 5Oxford Translational Myeloma Centre, NDORMS, University of Oxford, Oxford OX3 7LD, UK

**Keywords:** cancer care challenges, therapeutic advances, lung cancer, multiple myeloma, Hodgkin lymphoma, Latin America

## Abstract

**Simple Summary:**

The first Onco Summit: The Latin American (LATAM) Chapter was held from 19–20 May 2023, in Rio de Janeiro, Brazil. This educational initiative aimed to discuss the current challenges in cancer care, both globally and in LATAM, and foster the exchange of best practices to address these challenges. The Summit brought together more than 30 international and regional experts and more than 300 oncology specialists to deliberate on the advancements and barriers in oncology, with a focus on strategies tailored to the specific needs of LATAM. This report provides an overview of the key discussions among cancer specialists at the Summit.

**Abstract:**

The Onco Summit 2023: The Latin American (LATAM) Chapter took place over two days, from 19–20 May 2023, in Brazil. The event aimed to share the latest updates across various oncology disciplines, address critical clinical challenges, and exchange best practices to ensure optimal patient treatment. More than 30 international and regional speakers and more than 300 oncology specialists participated in the Summit. The Summit discussions centered on common challenges and therapeutic advances in cancer care, with a specific focus on the unique obstacles faced in LATAM and examples of adaptable strategies to address these challenges. The Summit also facilitated the establishment of a network of oncologists, hematologists, and scientists in LATAM, enabling collaboration to improve cancer care, both in this region and globally, through drug development and clinical research. This report summarizes the key discussions from the Summit for the global and LATAM oncology community.

## 1. Introduction

Cancer burden and associated mortality present a significant challenge to cancer care delivery worldwide, with a disproportionately higher impact on low-to-middle-income countries (LMICs) compared with high-income countries (HICs) [1]. By 2040, new cancer cases are estimated to increase by 95% in countries with a low Human Development Index (HDI), compared with 32% in countries with a very high HDI [1]. Cancer-related mortality is estimated to be higher in LMICs than in HICs, partly due to a lack of high-quality cancer care in LMICs influenced by their resource and access limitations [1,2]. In LMICs, cancer care is hindered by inadequate governmental incentives, restricted access to cancer medicines, insufficiently trained professionals, and the financial burden of diagnosis and treatment [2]. Although immunotherapy and targeted therapies are the gold standard treatment for advanced disease, the associated costs are substantial and present a significant barrier to their adoption in LMICs [2,3,4]. Efforts to address the increasing global cancer burden are marked by rapid development in new cancer treatments and advances in tumor immunology, leading to a surge in cancer immunotherapies. While these new treatment options hold promise, they also present challenges to clinicians in staying informed about the latest developments, determining the “best” choice and sequence of treatments in different lines and settings, and effectively implementing them in clinical practice to meet individual patient needs. This highlights the potential benefit of involving physicians to guide oncology-based therapeutic regimens and policies based on their clinical experience.

The challenges related to therapeutic advances in cancer are further complicated by significant disparities in cancer care and research between HICs and LMICs [2]. Current cancer research predominantly focuses on HICs, the findings of which may not be directly applicable to LMICs due to differences in challenges faced, disease characteristics, health policies, biological factors, geography, environmental conditions, resources, and treatment access and affordability. Additionally, only a small number of patients from LMICs participate in clinical trials. Therefore, LMICs need to conduct their own research in cancer care. This research should focus on addressing local and regional cancer needs, enhancing the quality of clinical care to ensure it is effective, feasible, and implementable, and supporting the development and retention of highly qualified cancer professionals [2,3,4].

In Latin America (LATAM), the burden of cancer is particularly pronounced, with a projected 67% increase in new cases by 2040 [5]. The increasing cancer burden in LATAM is aggravated by inadequate cancer care and preventive services, significant healthcare system disparities, limited resources, local barriers to the approval of novel agents, delayed diagnoses, and a lack of localized clinical trial data to guide personalized treatment [5,6]. The scarcity of national cancer plans, legislation for cancer prevention, and evidence-based cancer care strategies further complicates cancer care and research in the region [7]. Educational and awareness programs are crucial for tackling these challenges associated with the cancer burden, globally and specifically in LATAM. The Onco Summit 2023: The LATAM Chapter was an educational initiative sponsored by Takeda and held in Brazil to discuss some of the global and LATAM-specific challenges associated with cancer care and share best practices for addressing these challenges. This report summarizes the key discussions from the Summit on recent developments and challenges in oncology clinical practice and research.

## 2. Overview of The Onco Summit 2023: The LATAM Chapter

The Onco Summit 2023: The LATAM Chapter was an educational event organized by Takeda and held from 19–20 May 2023, in Rio de Janeiro, Brazil. The Summit aimed to foster international and local collaborations among clinicians, with a focus on sharing the most recent and unbiased updates in the different fields of oncology, addressing clinical challenges, sharing best practices, understanding the future of oncology, and working together to ensure the best treatment for all patients. More than 300 oncology specialists attended the Summit, including specialists in lung cancer, hematology, pathology, and diagnostics, enabling learning and interactions across many different fields of expertise. In addition, more than 30 speakers from around the world participated, with the majority being from LATAM, the United States, and Spain (Figure 1).

## 3. Recent Therapeutic Advances in Oncology

The opening session of the Summit featured a comprehensive discussion on the recent therapeutic advancements in oncology. The advancements in immuno-oncology have transformed the therapeutic landscape by introducing innovative treatments such as immunotherapies, radioimmunotherapy drugs, antibody–drug conjugates, and bispecific antibody therapies for cancer [8,9]. Activation of both the innate (innate immunity enhancers) and adaptive (redirected immunity approaches) immune systems can enhance immune surveillance, potentially improving identification and killing of cancer cells [10,11,12]. Therefore, it is crucial to focus on the innate and adaptive components of the immune system while also incorporating the foundational principles of the immune system into the development of all anti-cancer immune therapies. Oncolytic virus vaccines are another innovation in anti-cancer therapy specifically targeting tumor cells and able to code for therapeutic transgenes that can be selectively expressed in the tumor microenvironment (TME), enabling the TME to be conducive to cell death, and, therefore, triggering an antitumor immune response through tumor cell killing [13]. Significant advancements have also been made in novel targeted therapies for cancer, with several drugs available for different molecular abnormalities [14,15,16,17,18,19,20,21].

## 4. Common Challenges with the Use of Novel Anti-Cancer Therapies and Potential Solutions

The introduction of new treatment approaches is promising; however, it also presents a major challenge in terms of recruiting enough patients for timely testing. Additionally, ensuring that the trials are based on a pre-clinical rationale and that the enrolled patients are representative of the population is crucial. At the Summit, the challenges associated with using these novel treatments for lung cancer, multiple myeloma [MM], and lymphomas were discussed, along with the strategies to address them.

### 4.1. Lung Cancer

Immunotherapies and targeted therapies are commonly used in the first- and second-line treatment of non-small cell lung cancer (NSCLC). The challenges associated with their use include patient and treatment selection based on biomarkers, less defined biomarkers, overcoming resistance, lack of defined post-PD-L1 antibody treatment strategies, combination therapy decisions, reducing toxicity and improving quality of life, cure rates and survival, preventing CNS metastasis, unclear role of the combination of checkpoint inhibitors or different immunotherapies, and the need for new treatment strategies [22]. Similarly, although several targeted therapies are available for NSCLC, there are several barriers to their implementation in clinical practice, even in HICs, including early detection (screening), multidisciplinary care, and treatment costs [23].

#### 4.1.1. Limited Treatment Options Post-PD-L1 Antibodies in NSCLC

Newer immunotherapeutic approaches, such as alternative checkpoint inhibitors, immunostimulatory agents, bispecific T-cell engagers, tumor-infiltrating lymphocytes (TIL) therapy, chimeric antigen receptor (CAR) T-cell therapy, and vaccines, may offer additional treatment options following PD-L1 antibody therapy and are under investigation [22,24]. In addition to PD-1, the most commonly investigated immune checkpoint inhibitor in lung cancer is the lymphocyte activation gene-3 (LAG-3) inhibitor [22,24]. A Phase II trial of the soluble LAG-3 protein eftilagimod alpha and pembrolizumab in PD-1 unselected NSCLC showed an overall response rate of 42%, with two patients discontinuing the treatment [25]. Immunostimulatory agents targeting stimulatory pathways such as CD40 (e.g., APX005M), OX40 (e.g., MOXR0916, ivuxolimab), and 4-1BB (e.g., utomilumab) are under investigation [26,27,28]. Bispecific T-cell engagers (e.g., tarlatamab) are antibodies that target an antigen on the surface of the cancer cell while linking to the T-cell, causing T-cell activation [29,30]. TIL therapy has been reported to be efficacious against NSCLC in clinical trials. However, its use is limited by several factors, including the need for surgery to obtain TILs, the requirement for a dedicated facility, the lengthy manufacturing process, the need for high-dose interleukin therapy and its associated toxicity, and the intravenous administration, among others [31,32]. Mesothelin-targeted CAR T-cell therapy, in combination with pembrolizumab, is locally administered into the pleural space; however, the application and feasibility of this treatment are limited by the incidences of cytokine release syndrome (CRS) [33], a complex and lengthy manufacturing process, use after multiple lines of treatment as they result in selecting for more indolent disease, inpatient administration requirements, affordability, etc. [34,35]. Chimeric antigen receptor-engineered natural killer (CAR-NK) cells, currently in trials, are highly specialized cells that induce cell death via antibody-dependent cellular cytotoxicity and do not induce graft-versus-host disease (GVHD), allowing for allogeneic administration, but lack in vivo persistence and are highly susceptible to the immunosuppressive tumor microenvironment [36]. Future developments in CAR T-cell therapies for NSCLC should focus on developing patient tumor-specific NK and macrophage CAR constructs, combining therapies with customized tumor-specific antibodies, and exploring the use of anti-inhibitory or pro-stimulatory antibodies to improve treatment efficacy [37,38,39]. Vaccines for NSCLC have shown moderate success; however, these may be improved by the identification of new targetable antigens, adjuvant treatments, combining the antigen present in the tumor with a highly immunogenic construct, or combining vaccines with checkpoint inhibitors, immunostimulatory strategies, or cellular therapy [40,41]. The ideal future approach to systemic anti-cancer treatment is using a combination of synergistic immune-oncology agents to have robust anti-cancer effects [8,9].

#### 4.1.2. Selecting Targeted Therapy Based on Biomarker Selection

Biomarker testing is key in selecting an optimal targeted therapy for each patient in the neoadjuvant and adjuvant settings [42]. All patients with advanced-stage NSCLC should be tested for both epidermal growth factor receptor (EGFR) mutations and anaplastic lymphoma kinase (ALK) rearrangements, the most common forms of NSCLC, before initiation of first-line treatment, with a maximum turnaround time of 10 working days [43,44,45]. Additionally, using the concepts of drug structure, preclinical models in pathology reports may help clinical oncologists choose the right targeted therapies. If a targeted therapy is active against a specific mutation, it does not guarantee effectiveness against other mutations. This is due to varied responses of mutations to different targeted therapies (e.g., EGFR inhibitor, osimertinib) [46,47,48,49]. The selection of the right choice of targeted therapies needs to be based on their efficacy and safety profile, as well as patient requirements (prevention of brain metastases, local regulations, etc.) [23]. Therefore, oncologists must stay up to date on the mechanisms of action of different targeted therapies, comparative clinical trial data, patient profiles suitable for each type of targeted therapy, etc., to choose the right treatment option for their patients. Evidence suggests the combination of EGFR/ALK tyrosine kinase inhibitors (TKIs) with checkpoint inhibitors does not significantly improve progression-free survival (PFS) or survival rates [50]. Patients with ALK rearrangement in NSCLC and small or asymptomatic brain metastases should be treated upfront with newer generation TKIs (e.g., brigatinib, alectinib, ceritinib, lorlatinib) [51,52].

#### 4.1.3. Treatment Resistance to Targeted Therapies

Treatment resistance in NSCLC could be intrinsic or acquired, involve complex mechanisms, and can be influenced by TP53 co-mutations and other clinical characteristics [23]. Intrinsic resistance can be minimized by optimizing the kinase inhibitor, e.g., osimertinib. The next generation of targeted therapies can address on-target resistance but may also pose the risk of developing clinically challenging off-target resistance mechanisms [23]. Future treatment strategies to overcome resistance to targeted therapies should consider combination therapies in the first-, second-, or later-line settings, patient monitoring using circulating tumor DNA, strategies to address co-mutations, etc. [53]. Tumor and liquid biopsies to evaluate the genomic events at the time of acquired resistance can help choose appropriate subsequent or rescue treatments [23].

#### 4.1.4. Key Takeaways to Address the Challenges in Lung Cancer

Future clinical trials in NSCLC should focus on exploring novel treatment strategies, particularly in second-line settings where resistance to first-line agents has emerged. Additionally, investigating the optimal sequencing of treatments remains crucial.

In resource-limited countries, such as those in LATAM, efforts need to be made to improve access to biomarker testing, adequate tissue diagnostics, surgery, and newer neoadjuvant or adjuvant treatment strategies to implement targeted therapies in patients with NSCLC [54,55].

Besides improving therapies, there is also a need to focus on preventing lung cancer by supporting policies such as encouraging tobacco cessation in smokers, reducing air pollution, limiting occupational exposure to radon gas, etc. [56].

### 4.2. MM

The treatment of MM is complex, and common treatments include immunomodulatory agents, proteasome inhibitors, stem cell transplantation (SCT), or immunotherapy [57,58]. Treatment decisions at initial diagnosis and relapse are based on patient factors (age, frailty, comorbidities, performance status, preference, and affordability), prior therapy (response and toxicity to prior therapy, prior SCT, time from prior therapy, and availability of clinical trials), disease burden (duration of prior remission, symptomatic versus asymptomatic, and cytogenetics), and availability of and access to treatment [57,58]. Despite the availability of several novel agents, the treatment of patients with relapsed/refractory MM (RRMM) is clinically challenging due to clonal evolution, refractoriness to treatments, poor quality of life, and cumulative end organ damage [57,58,59,60,61]. The availability of several novel treatments has made the choice and sequencing of the correct treatment regimens in RRMM complex and challenging [57,58]. Although randomized controlled trials (RCTs) provide evidence of the efficacy and safety of these novel agents, evidence suggests discrepancies between clinical trial data and real-world use of these drugs. These are due to selection bias in trials arising from eligibility criteria, study design, clinical trial settings, fixed durations of treatments, and rigorous protocols [62]. Prior exposure to lenalidomide presents another major challenge when determining the optimal treatment for patients with RRMM, as a substantial proportion have been exposed to lenalidomide during frontline therapy. Crucially, lenalidomide refractory patients have been shown to have worse outcomes when treated with lenalidomide-based salvage or maintenance therapies compared with patients who previously responded to lenalidomide treatment [63,64]. Moreover, disease relapse in myeloma can be driven by subclones, which may display sensitivity to prior therapies [64]. Selecting or identifying the type of patients with prior lenalidomide exposure who may or may not benefit from optimization with a lenalidomide-based combination is a clinical challenge due to the absence of a biological construct to define absolute lenalidomide-refractoriness [65]. Additionally, improving maintenance therapy and continuous therapy [57,58], treatment of elderly patients [66,67], and effectively managing smoldering myeloma [68] are other clinically significant challenges in MM.

#### 4.2.1. Making Treatment Decisions in RRMM Solely Based on Clinical Trial Data

Real-world evidence (RWE) provides valuable insights into the effectiveness and safety of treatments in real-world clinical settings, complementing the evidence obtained from RCTs [69,70]. RWE in RRMM, particularly in combinations involving lenalidomide, has shown significant differences in PFS outcomes when compared with RCTs for combinations such as daratumumab/lenalidomide/dexamethasone [71,72] and carfilzomib/lenalidomide/dexamethasone [73,74,75]. Additionally, RWE suggests that lenalidomide-based combination treatments have inferior outcomes, compared with RCTs, in patients with prior lenalidomide exposure. However, ixazomib/lenalidomide/dexamethasone combination datasets have shown consistent outcomes [76,77,78]. By contrast, real-world outcomes of non-lenalidomide-based combination treatments were inferior compared with their RCT data and compared with lenalidomide-based therapies [73,79,80]. Therefore, combining RCT and RWE provides a comprehensive and complementary approach to generating clinical evidence for the management of RRMM [69,70].

#### 4.2.2. Treatment of Patients with RRMM with Prior Exposure to Lenalidomide

Patients with prior exposure to lenalidomide can be characterized based on the following criteria: (1) prior response or refractoriness to lenalidomide; (2) assessment of refractory, relapsed, or both to lenalidomide with baseline molecular genetic testing to select patients with high-risk genetic features who are unlikely to benefit from lenalidomide and, thus, should not be re-exposed to lenalidomide-based therapies; (3) duration, dose, and compliance on prior treatment with lenalidomide to select patients who may be lenalidomide-sensitive; and (4) the reason for discontinuation of lenalidomide (not treated until progressive disease, discontinuation for other reasons, adverse events (AE) on partner drugs, and indolent relapse) [66,81,82]. Switching drug class using a combination approach and/or switching to a second-/third-generation agent in the same drug class (e.g., switching from lenalidomide to pomalidomide) is often the preferred option in second-line treatment [63,83]. Re-treatment with lenalidomide can be considered if the treatment initially produced a clinically meaningful response without unacceptable toxicity [63,83]. For patients with prior treatment with bortezomib/lenalidomide or daratumumab/lenalidomide combination, lenalidomide-based regimens (including ixazomib/lenalidomide) are recommended for patients who continue to be lenalidomide-sensitive [63,83]. Studies (TOURMALINE-MM1) conducted in a large cohort of relapsed myeloma patients with prior lenalidomide exposure support the consideration for the use of ixazomib/lenalidomide at relapse in lenalidomide-sensitive patients [77,78]. Re-treatment with higher doses in later lines may be considered in patients who may become sensitive to (escalated dosages of) drugs to which they were previously refractory, as demonstrated by the appearance of different tumor clones during subsequent stages of the disease. Lenalidomide-free regimens could also be considered at relapse for patients with prior lenalidomide exposure, but with no substantial advantage in PFS in reported data [76]. Further data from ongoing and future trials, with molecular stratification, are required to differentiate lenalidomide-refractory patients who may or may not benefit from subsequent lenalidomide-based therapies.

#### 4.2.3. Selecting the Type and Duration of Maintenance and Continuous Therapies

Lenalidomide is the standard of care in several countries for maintenance and continuous therapy in patients with RRMM who are eligible and ineligible for SCT, respectively. Most patients can be maintained on lenalidomide if it is well tolerated. Ixazomib, an oral medication, is a viable alternative for those who cannot tolerate lenalidomide [57,58]. Double maintenance may be needed in patients with high-risk cytogenetics rather than lenalidomide maintenance to prevent resistance to lenalidomide, and the common agents used are carfilzomib/lenalidomide/dexamethasone or ixazomib/lenalidomide/dexamethasone. High doses of cyclophosphamide for maintenance therapy should be avoided due to the risk of inducing mutations; however, metronomic doses of cyclophosphamide may be used in combination with other treatments to achieve clinical benefit. Immunotherapy may also be considered for maintenance and continuous therapies and is currently under investigation. Determining the duration of maintenance therapies and establishing the criteria to define the fixed duration of maintenance therapy pose clinical challenges, as maintenance therapy cannot be administered continuously and practices vary widely across different clinical settings. Trial data are required to define a fixed duration for maintenance therapies either based on minimum residual disease (MRD), baseline genomics, or time period. The use of MRD negativity to guide the duration of maintenance therapies is a useful option, and several trials in Europe are trying to assess the fixed duration of maintenance therapies based on MRD negativity. This may become feasible in the future once MRD tests transition to blood-based formats.

#### 4.2.4. Treatment of Elderly Patients

In elderly patients, the dosing of lenalidomide or ixazomib may require adjustment of the starting dose compared with those used in clinical trials. Currently, there are no prospective trial data available to guide the dosing of lenalidomide or ixazomib treatment in elderly patients [66]. For very elderly patients with renal failure, the use of an ixazomib/lenalidomide/dexamethasone combination for therapy should start at a lower dose based on the creatinine clearance rate [66]. However, if the patient’s condition improves, the dose may be increased to achieve a deeper response. Since myeloma patients receive long-term therapy, they usually require dose modification over their course of treatment based on AEs and their clinical condition. Commonly used dose reductions in clinical practice are 2.3–3 mg for ixazomib, 10 mg for lenalidomide, and 4 mg for dexamethasone. Carfilzomib may be used in all elderly patients with cardiac arrhythmia; however, they should be started at lower doses and baseline cardiac tests should be performed [67]. Starting carfilzomib at higher doses in these patients may eventually lead to cessation of the treatment due to its cardiac AEs. Therefore, rather than excluding its use, it may be better to start at lower doses to achieve the desired response in these patients [67].

#### 4.2.5. Treatment of Smoldering Myeloma

Managing patients with high-risk smoldering myeloma is challenging, as current recommendations in most countries suggest treatment solely within clinical trials. However, the prospect of future treatment options is promising and dependent on establishing appropriate endpoints for smoldering myeloma and the availability of effective drugs to ensure optimal clinical outcomes. Notably, the treatment for patients with high-risk smoldering myeloma will differ across patients, with some patients receiving intensive treatment while others may be managed with monotherapy. Lenalidomide, frequently used as maintenance therapy for newly diagnosed high-risk smoldering myeloma, has been reported to significantly delay progression to symptomatic multiple myeloma and improve outcomes [68]. Additionally, combination therapies involving ixazomib have been shown to be effective and well tolerated in high-risk smoldering myeloma [84].

#### 4.2.6. Key Takeaways to Address the Challenges in MM

Incorporating RWE into the decision-making processes for the treatment of RRMM can enhance the understanding of treatment outcomes in routine clinical practice, which may, in turn, help with licensing and reimbursement decisions in RRMM [69,70].

When managing patients with prior exposure to lenalidomide, the treatment approach should consider their prior response to lenalidomide, relapse status, or refractoriness. Strategies may involve switching drug classes, using combination therapies, transitioning to second-/third-generation agents within the same drug class, or re-treatment with lenalidomide or lenalidomide-based regimens in those with lenalidomide sensitivity.

The choice of maintenance therapy in RRMM involves considering factors such as administration route, efficacy, safety, tolerability, and impact on quality of life [57,58]. Commonly used maintenance therapies include lenalidomide and ixazomib.

In elderly patients with RRMM, the dosing of novel agents should be based on frailty tests, renal function, bioavailability data, and RWE to optimize their treatment plans [66].

In LATAM, most patients with MM are treated in the public healthcare sector. The challenges faced by these patients include delayed diagnosis, limited access to diagnostic tests such as molecular and cytogenetic testing, delayed autologous stem cell transplantation (ASCT), limited access to novel treatments, and disparities in care between the public and private healthcare sectors [85,86,87].

Increased awareness of MM, early diagnosis, accelerated approvals, and improved access to novel diagnostics and novel treatments in the public healthcare sectors. Increased access to ASCT, participation in local clinical trials, etc. are some strategies to improve the management of MM in LATAM.

### 4.3. Lymphomas

During the Summit, Hodgkin lymphoma (HL) and T-cell lymphomas were discussed. Although HL has a cure rate of 70–80% in advanced-stage disease, the management of patients with relapsed/refractory HL (RRHL) remains challenging [88,89,90,91,92]. High-dose chemotherapy followed by ASCT is the standard of care in RRHL [43,44]. However, a considerable proportion of patients (up to 50%) experience relapse after ASCT [88,89,90]. Moreover, patients who are ineligible to undergo ASCT have limited treatment options [89,90,92].

T-cell lymphomas constitute a heterogenous group of malignancies [93,94,95]. A lack of clear risk-adapted treatment strategies and specialized treatment in high-risk patients, inadequate response rates to available therapies, etc., are some of the major challenges in the treatment of T-cell lymphomas [93].

#### 4.3.1. Improving SCT Rates and Clinical Outcomes with ASCT in RRHL

The main goals of treatment in ASCT-eligible patients with RRHL are to improve chemosensitivity and achieve a complete metabolic response, enable stem cell mobilization, prevent further relapse, and eventually achieve a cure [89]. Both pre- and post-transplant strategies impact outcomes with ASCT. In the pre-transplant phase, the first step is improving the eligibility rate for ASCT through optimized frontline and salvage treatment and bridging therapy to SCT. Tailoring salvage treatment based on PET scans and metabolic tumor volume assessment before ASCT can improve clinical outcomes after ASCT [89,96]. Pre-transplant strategies that incorporate targeted therapy or anti-PD-1 agents may improve response rates and eligibility for ASCT [89,91,92,96,97,98,99]. Real-world data in high-risk patients with RRHL (refractory to two or more consecutive systemic therapies) have shown that ASCT after anti-PD-1 therapy was associated with favorable outcomes, even among heavily pre-treated, previously chemorefractory patients [100]. Prior to ASCT, the use of novel agents, such as brentuximab vedotin, in frontline settings, salvage treatment, and as a bridge to ASCT, has been shown to improve chemosensitivity and clinical outcomes [89,91,96,97,98]. Pembrolizumab plus gemcitabine/vinorelbine/liposomal doxorubicin as a pre-transplant salvage regimen showed favorable response rates and was well tolerated in patients with RRHL [101]. In the post-transplant settings, consolidation and maintenance therapies, tandem SCT, and post-ASCT radiotherapy may improve post-SCT outcomes [96,99]. Consolidation therapy is the most common approach, and consolidation with brentuximab vedotin has been shown to prevent further relapse and use of subsequent therapies following ASCT, especially in high-risk, heavily pre-treated patients [92,96,97,98,99,100]. Additionally, prospective clinical trials suggest that consolidation with checkpoint inhibitors alone or in combination with brentuximab vedotin may be feasible, well tolerated, and effective [89,102,103]. In patients with RRHL who experienced ASCT failure, immunotherapy and targeted therapies have shown favorable clinical outcomes [89,96,104]. Brentuximab vedotin monotherapy and anti-PD-1 agents, either as monotherapy or in combination, may improve response rates in these patients. Other options, such as triplet combinations of brentuximab vedotin and nivolumab with CAR T-cells and other targeted agents (ipilimumab, anti-CD30/CD16A-bispecific antibody, etc.), have demonstrated favorable preliminary results and are being further investigated as treatment options. Allogenic SCT is still a feasible option in these patients despite long-term toxicity, and its role in the era of new anti-cancer agents is being revisited [89,96,104].

#### 4.3.2. Treatment of SCT-Ineligible Patients with RRHL

The main treatment goals in ASCT-ineligible patients with RRHL are to achieve and maintain disease control, minimize morbidity, and retain quality of life [105,106]. In these patients, there may be a potential for cure in a minority of cases in the second-line setting. However, their chances of cure decrease with later lines and are minimal with only conventional chemotherapy [105,106]. Administering targeted therapy or immunotherapy to these patients early in the treatment cycle, especially in salvage settings, should be considered to achieve the targeted goals. Brentuximab vedotin monotherapy and anti-PD-1 agents or their combinations are available options, as they have shown favorable outcomes and improved SCT eligibility in SCT-ineligible patients and those unsuitable for multi-agent chemotherapy [97,98,105,106,107,108]. Brentuximab vedotin may act as a bridge to SCT in patients with RRHL who achieve a suboptimal response to multi-agent frontline/salvage chemotherapy or radiotherapy [98,107,108].

#### 4.3.3. Treatment of T-Cell Lymphoma

Correct diagnosis, staging, and subtyping are the fundamental pillars in decision making in T-cell lymphomas, which translate into therapeutic success [94,95]. Treatment strategies should be tailored based on the patient’s needs and their individual risk, as well as the convenience of administration and comparative efficacy [93]. While no single drug is perfect for treating these pathologies [93], using available drugs in appropriate combinations may offer significant benefits. Novel agents such as brentuximab vedotin have shown efficacy in treating malignancies, including cutaneous lymphomas, and should be further explored [94]. A deeper understanding of T-cell biology and disease progression, coupled with advancements in new therapeutic platforms, should inform the development of innovative treatment strategies for peripheral T-cell lymphomas [93]. Since peripheral T-cell lymphomas are considered orphan diseases due to their rarity, the development of newer therapies and clinical trials should be pursued collaboratively.

#### 4.3.4. Key Takeaways to Address the Challenges in Lymphomas

Recent advances in immunotherapy have aimed to address challenges associated with managing various lymphomas with the use of agents such as brentuximab, vedotin, and immune checkpoint inhibitors (nivolumab, pembrolizumab) either as monotherapy or in combination with chemotherapy [89,92].

ASCT remains the cornerstone of RRHL treatment. Strategies pre- and post-ASCT should aim to improve chemosensitivity and complete response rates, which can, in turn, positively impact ASCT, improve outcomes, and reduce post-ASCT relapse [89].

Interpreting clinical trial data along with RWE is essential for guiding treatment decisions in RRHL [109,110]. Results from RWE studies should be considered within the context of the individual study design, as well as the inherent and study-specific limitations of RWE [109,110].

Managing lymphomas is challenging, involving complex treatment decisions that require a thorough understanding of patient characteristics, their treatment goals, and available novel treatment options [89,92]. This can be achieved only through collaboration among specialists, emphasizing the significance of a multidisciplinary approach to ensure comprehensive and coordinated care for patients with RRHL [89,92].

## 5. Challenges with Therapeutic Advances in Cancer—Focus on Oncology Clinical Trials and Drug Development Process

One of the main challenges with the advancement of cancer therapies is conducting clinical trials that are clinically relevant and applicable globally, as well as translating them into clinical practice (Table 1). Current oncology clinical trials are limited by several factors, including the under-representation of certain high-risk groups, ethnicities, and geographies; inappropriate control arms, especially for trials conducted in LMICs; a lack of clinically relevant endpoints; inclusion of only surrogate endpoints; limited RWE to confirm clinical trial data; and a lack of follow-up data [111,112,113,114]. LATAM and several other resource-limited regions are under-represented in oncology clinical trials due to several challenges, including a limited number of oncology trials in these regions, differences in public and private healthcare sectors, inclusion criteria not suitable for local population needs, and competing clinical priorities [111,113]. Another challenge with the advancement of cancer therapies is the drug discovery and development process, which is complex and has several problems, including patent cliffs, the geopolitical situation, a lack of personnel with a trained skillset, inflation, limited research and development success, and long approval processes [115,116,117,118]. While a patent cliff can allow the drug to reach more patients, it erodes the revenue of pharmaceutical industries, rendering less money for future research and development investments [119]. There is an increased number of new molecules entering preclinical research in oncology, but their success rate is low. Additionally, although the average time for getting the drug to the market has reduced, the number of drugs approved is fewer due to consistent failures in different phases of clinical trials [115,116,117,118,120]. Moreover, the success rates of the approved drugs are minimal [115,116,117,118]. To address these challenges, the industry research and development processes try to focus on complex therapies, resulting in limited innovative therapies from the industry.

### 5.1. Improving Generalizability and Applicability of Clinical Trial Data in Clinical Practice

Clinical trial designs should be improved by expanding the inclusion criteria to suit different ethnicities, geographies, and patient groups [111,112,113,114]. Clinical trials should include high-risk patients, patients with unmet needs, and patients from LMICs. It is important to ensure that appropriate control arms are included for reimbursement approvals and that clinically relevant and evolving endpoints are chosen based on the type of trial, disease, and approvals involved. In some cases, it may be better to accept clinical trials with surrogate endpoints if the drug may benefit patients rather than waiting for years for survival data. However, when using surrogate endpoints, it is crucial to select clinically meaningful endpoints with the condition that long-term survival data will be provided in due course [111,112,113,114]. Additionally, the use of RWE [70,121] and digital solutions for remote patient monitoring, AE monitoring, patient diagnosis, etc. [122] are critical for improving the future of oncology care. Clinicians should advocate for RWE trials to the government and payers by emphasizing their importance, benefits, and impact on clinical practice to obtain funding for high quality real-word data generation. Finally, publication bias should be avoided when reporting trial results. Regular update reports of trial progress and results, including negative results, should be emphasized, and transparent reporting should be ensured [123]. The under-representation of LMICs in cancer clinical trials may be overcome by influencing and negotiating with the clinical trial sponsors to design clinical trials based on local population needs. However, sponsors will need to be convinced that the clinical trial results will be translated into local clinical practice through local regulatory and reimbursement approvals. The success of one local clinical trial being implemented into clinical practice can influence subsequent clinical trials to be designed to suit local needs. Identifying the most important unmet needs in patients according to clinical physicians, crystallizing those unmet needs, and involving important stakeholders who can work towards addressing those needs can improve the quality of the design of local clinical trials. Finally, reimbursement policies influence the advancement of anti-cancer therapies. When several different products are available for the treatment of the same condition in oncology, payers and regulatory bodies will need to assess the most effective agent in their patient populations before approving or subsidizing them.

### 5.2. Improve the Drug Development Process in Oncology

One of the most common strategies adopted by the pharmaceutical industry to overcome the challenges of the drug development process and launch new drugs at a faster and cost-effective rate is the open innovation method [124,125]. With this method, some of the strategies used involve understanding market trends, restructuring to consolidate research and development to focus on selected areas and decrease investment in other areas, outsourcing the work at different stages of clinical development, and opening innovation platforms. The latter allows industries to buy, sell, or collaborate with academia, venture capitalists, small biotech companies, etc., to bring new molecules to the market, with increased chances of approval, and allows innovators to utilize their infrastructure and platform to innovate their product by keeping their intellectual property rights with them [124,125].

### 5.3. Key Takeaways to Address the Challenges with Clinical Trials and the Drug Development Process

Clinical trials in oncology should prioritize enhancing generalizability and clinical relevance. This can be achieved by including patients from LMICs with unmet needs, ensuring appropriate control arms for reimbursement approvals, and including clinically meaningful, evolving endpoints based on the type of trial, disease, and regulatory requirements.

To advance cancer therapies, the healthcare sector, particularly regulators, should focus on establishing frameworks for accelerated drug discovery, facilitating early access programs, and expediting marketing authorization.

Awareness of the challenges in drug development is the key to fostering future collaborations and improving innovative engagements within the pharmaceutical industry.

Advocacy, discovery, and political ambition are important for the advancement of cancer therapies.

## 6. Challenges with Cancer Care in LATAM

In LATAM, significant challenges limit diagnostic and therapeutic advances compared with Western countries (Figure 2) [126,127,128]. Most patients in LATAM countries receive treatment from the public healthcare system, with only a small proportion receiving care from private healthcare. There is a huge disparity between the public and private healthcare systems in the cancer care provided. The public healthcare system lacks access to modern diagnostic techniques and innovative treatments, leading to relatively inferior clinical outcomes. There is no equitable access to even well-established therapies, such as ASCT for lymphomas and MM, in some LATAM countries, specifically through public healthcare. An exception is seen in Colombia, where the gap between public and private healthcare is relatively nonexistent due to the subsidized healthcare policy. Access to diagnostic tests needed for treatment decisions, progress monitoring, and maintenance therapy planning—ranging from basic diagnostics to molecular testing—is often limited, leading to delayed diagnosis and delayed referral to ASCT. Further, access to novel therapies such as immunotherapies or targeted therapies is usually limited due to delayed approval or incorporation, mainly in public institutions. As a result, very few patients receive novel treatments as part of their frontline regimens. As a result, there is a disparity in outcomes and survival between public and private healthcare facilities [86,126,127,128]. Studies from LATAM have shown that the increased use of novel agents over time was higher in private clinics compared with public clinics [86,129]. The access to care also varies vastly from country to country in this region [126,127,128]. The differences in treatment patterns by country and clinic type (private versus public) and poor clinical outcomes among patients with different cancer types indicate areas of unmet therapeutic needs in LATAM [86]. There are ongoing efforts in various LATAM countries to address these issues and establish equity between public and private healthcare systems (Table 2).

### 6.1. Improved and Equitable Access to Modern Diagnostic Techniques and Novel Therapies

A practical approach to addressing the issue of limited access in public healthcare systems is emphasizing the significance of early diagnosis and early treatment initiation and, therefore, ensuring advanced diagnostic tests and innovative treatments are made available to a maximum number of patients. Another way to address the limitations of restricted access to novel treatments in public healthcare is by selecting the patient types who will have the maximum benefit from novel treatment and administering novel agents to only those patients to reduce costs. Efforts to increase participation in clinical trials and increase local clinical trials that address the specific and most important clinical unmet needs of the local population in LATAM can improve access to newer treatment and diagnostics locally. The countries in this region should create policies for equitable access to basic and novel diagnostic and therapeutic interventions to improve survival through constant education programs combined with political pressure and the use of population-based cancer registries [5,6]. Additionally, frequent, relevant, and practical discussions on policy and pricing for novel agents with policymakers are imperative.

### 6.2. Early Diagnosis and Early Intervention

The awareness about different cancer types in the LATAM community, except among relevant experts in the field, is inadequate, which may be responsible for the delayed diagnosis in this region. Educational programs to increase awareness, including that of specialists in other fields, may be helpful. Additionally, educating the healthcare community on the significance of early diagnosis and early treatments may enable earlier cancer diagnosis. Facilitated access to medical examinations, from simple diagnostics to molecular testing, and an increased number of beds for ASCT may be some simple strategies to address this issue. The goals for early cancer management in LATAM healthcare systems must include clear and effective communication. This should include educating the public with up-to-date, validated evidence on cancer prevention and establishing a strong evidence base on cancer risks. These efforts will enable the development and adaptation of regional recommendations [6].

### 6.3. Key Takeaways to Address the Challenges with Cancer Care in LATAM

The LATAM oncology community faces several challenges, including limited access to basic and innovative diagnostic and therapeutic strategies, disparities between private and public healthcare sectors, delayed diagnosis and treatment, inadequate adoption of novel therapies in local clinical practice, and under-representation of local unmet needs and patients in clinical trials.

Addressing these challenges requires ongoing awareness and educational initiatives targeting healthcare professionals, government bodies, and payers.

Efforts should focus on early diagnosis and identifying patients most likely to benefit from novel treatments to reduce costs.

Other strategies to improve cancer care in LATAM include enhancing participation in clinical trials, maintaining appropriate registries and databases, and creating local and regional policies for subsidized healthcare and equitable access.

## 7. Conclusions

The Onco Summit 2023: The LATAM Chapter, successfully addressed prevalent challenges in oncology and facilitated the exchange of global best practices. By emphasizing recent therapeutic advances, especially immunotherapy and targeted therapies, the Summit shed light on overcoming common challenges in delivering optimal cancer care. It identified the key challenges across LATAM, shared universal unmet needs, showcased adaptable strategies, and identified collaboration opportunities for improving oncology care through drug development and clinical research. By enabling unbiased discussions among oncology professionals and showcasing the potential of global clinical trials in local clinical practice, the Summit proved pivotal for a region grappling with significant challenges and obstacles to cancer care. The event’s focus on international achievements and cutting-edge methods in oncology care and clinical research promises continuous quality improvement in treating diverse cancer types. Crucially, the Summit established a network of oncologists, hematologists, and scientists in LATAM countries, fostering collaboration towards the shared objective of advancing global and regional cancer care.

## Figures and Tables

**Figure 1 cancers-16-03063-f001:**
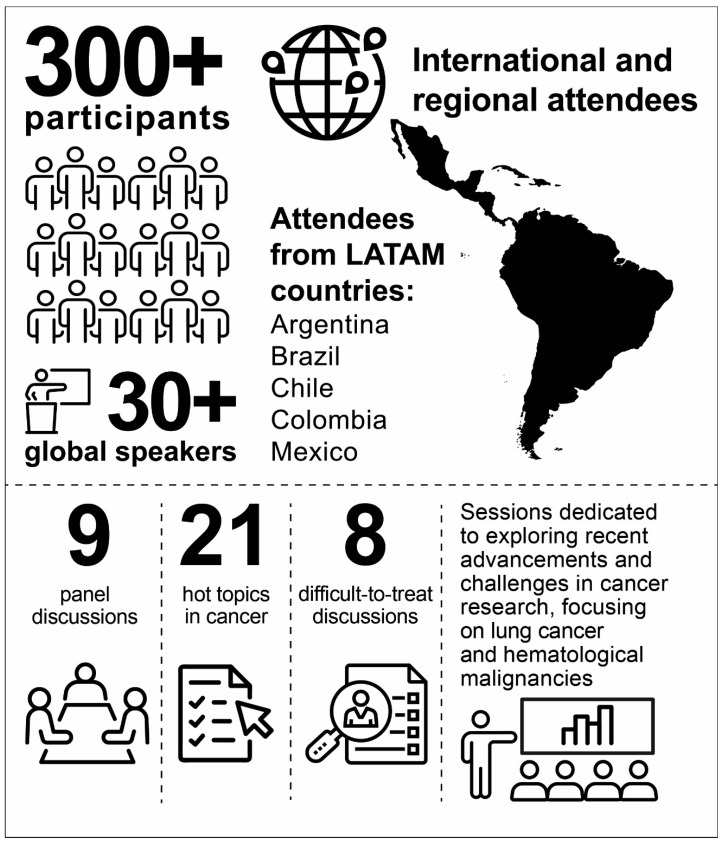
The Onco Summit 2023: The LATAM Chapter congress proceedings. More than 300 oncology specialists and over 30 international and regional speakers participated in the Summit. Most attendees were from Argentina, Brazil, Chile, Colombia, and Mexico. There were nine panel discussions, 21 hot topics in cancer, and eight difficult-to-treat discussions; all sessions were dedicated to exploring recent advancements and challenges in cancer research, with a focus on lung cancer and hematological malignancies. LATAM: Latin America.

**Figure 2 cancers-16-03063-f002:**
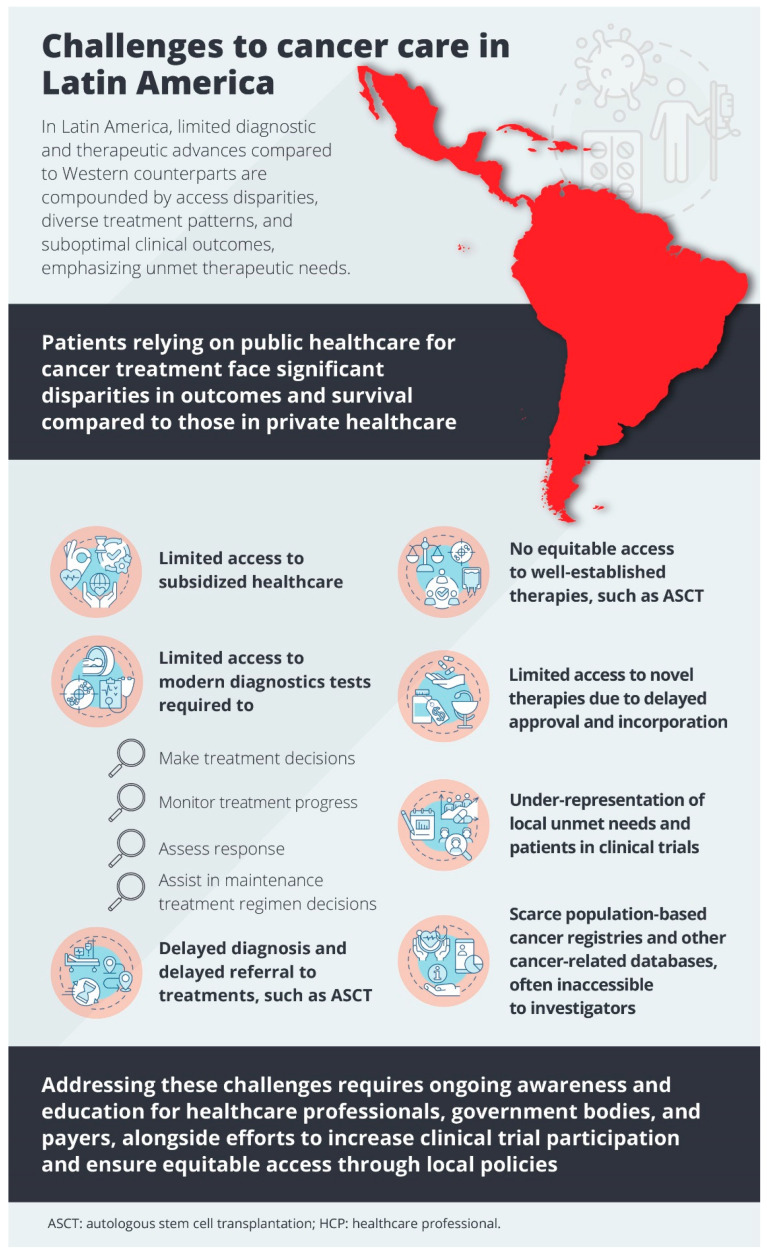
Challenges to cancer care in LATAM. Several challenges limit diagnostic and therapeutic advances in LATAM, including disparities in access to subsidized healthcare and modern diagnostic tests, diverse treatment patterns, and suboptimal clinical outcomes in patients relying on public healthcare compared to those in private healthcare. LATAM: Latin America.

**Table 1 cancers-16-03063-t001:** Common challenges with therapeutic advances in cancer.

Challenges	Potential Solutions
Clinical trials	
Under-representation of certain high-risk groups, ethnicities, geographies Inappropriate control arms Lack of clinically relevant endpoints Inclusion of only surrogate endpoints Limited RWE to confirm clinical trial data Lack of follow-up data	Include high-risk patients, patients with unmet needs, and patients from LMICs in clinical trials Ensure appropriate control arms for reimbursement approvals, and include clinically relevant and new and evolving endpoints Choose clinically meaningful surrogate endpoints, ensuring that long-term survival data will be provided later. Explore digital solutions for remote patient monitoring, AE tracking, and diagnosis. Advocate for more RWE studies
Under-representation of LMICs in cancer clinical trials	Negotiate with sponsors to tailor trials to local needs and boost local participation. Work with local regulatory and reimbursement authorities to translate trial data into clinical practice. Identify key unmet needs and involve all stakeholders in trial design.
Drug development process	
Complex drug development process Limited research and development success Long approval processes Low success rate of approved drugs	Consolidate research and development to focus on selected areas, outsource different stages of clinical development, and open innovation platforms for collaboration Develop routes for accelerated drug development, early access, and accelerated marketing authorization

AE: adverse event; LMIC: low-to-middle-income countries; RWE: real-world evidence.

**Table 2 cancers-16-03063-t002:** Challenges with cancer care specific to LATAM.

Challenges	Potential Solutions
Limited access to novel diagnostic techniques and treatments Delayed approval or incorporation of novel therapies	Emphasize early diagnosis and treatment initiation Target novel treatment to patient most likely to benefit can reduce treatment costs Increase local clinical trials and participation
Delayed diagnosis, delayed treatment initiation, and delayed referral for ASCT	Improve awareness of different cancer types and characterization Educate the healthcare community on the significance of early diagnosis and treatments Facilitate access to exams, from simple diagnostics to molecular testing, and increase number of beds for ASCT
Disparity between public and private healthcare systems	Develop policies for equitable access using regular education, political advocacy, and population-based cancer registries. Engage in regular, practical discussions on policy and pricing for novel agents.

ASCT: autologous stem cell treatment; LATAM: Latin America.

## Data Availability

No new data were created or analyzed in this study. Data sharing is not applicable to this article.

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
