# Peer review of "Proceedings from the First Onco Summit: LATAM Chapter, 19–20 May 2023, Rio de Janeiro, Brazil"

_cancers, 2024, doi:10.3390/cancers16173063_

Round 1
Reviewer 1 Report
Comments and Suggestions for Authors
Comments and suggestions to the authors of cancers-3120229 (Vania Hungria, Anna Sureda, Garcia Rosario Campelo, Marco Aurélio Salvino and Karthik Ramasamy. Proceedings from the First Onco Summit: LATAM Chapter, 2 May 19–20, 2023, Rio de Janeiro, Brazil)
Brief summary of the format and style: The manuscript summarize the key discussions held between cancer specialist on the first Onco Summit: The Latin American Chapter, which was held from May 19–20, 17 2023, in Rio de Janeiro, Brazil in 18 pages without references included 2 figures and 2 tables. The manuscript written in Cancers word template and format.
Suggestions for the authors:
Formal: The manuscript is well written, easy to read.
Some small changes might help the readers wich detailed bellow:
1. In the tables the content of the “Potential solution” would be better if just stated as a list with statements consist of some key word (3-5), instead of sentences. Complete sentences and explanation should be only in the main text.
2. Also, it would be better to link the “Challenges” in the first column to the short statements in the “Potential solution” either by using bullet point or not but to be lined up to each other. After these changes the style of the 2 tables should be the same (both with bullet point or both without them). Just so that the information content can be absorbed more easily and quickly.
Content:
1. In line 134: should insert after the “in clinical practice” the “even in the HICs”. Although this sentence is in the “common challenges” section, it would highlight that more these are challenges not only in LATAM.
2. In line 155: “space but is limited by…” I think between “but” and “is limited” is missing the words “its effectiveness” - as in my understanding the occurrence of subsequent CRS (?) limits its effectiveness. Or, if the meaning should be that “the pleural administration is limited because of the CRS” then fine in the recent state.
3. In line 188: before the “role” I think need the word “positive”, as the evidence of the positive role of combination is missing probably.
4. In line 191: To whom the adjective "selected" applies? Which type of patients? As this is not stated in the previous sentence as I have not seen? Should be defined clearly what type of selection, namely, except of what type of patients.
5. In line 194: instead of “co-mutations (TP53)” should be “co-mutations (i.e in TP53 genes)” if the meaning was planned as “influenced by co-mutation in TP53 gene”. Just to clarify that we know for sure if the commutation is with TP53 and other genes or in the same gene.
6. In line 210-212: Need references for this statement which support the necessity of these changes and would detail even more the necessary preventive fields if were discussed more in detail in the Summit.
7. In line 239-241: These statements require citations - even though they are explained in detail in the following sections.
Reviewer 2 Report
Comments and Suggestions for Authors
1. Original Submission
Recommendation to the author and editor:
Major revision
Title: Manuscript ID: cancers-3120229 entitled "Proceedings from the First Onco Summit: LATAM Chapter, 2 May 19–20, 2023, Rio de Janeiro, Brazil”
Article Type: article
2. Comments to the Corresponding Author:
COPE Ethical guidelines followed during the review process,
The manuscript addresses the The Onco Summit 2023: The Latin American Chapter was held in Brazil from May 19–20, 2023, spanning two days. Authors described the this event's primary goals and updates in various oncology fields, tackle critical clinical challenges, and exchange best practices for optimal patient treatment. According to author’s information, over 30 international and regional speakers and more than 300 oncology specialists attended the Summit. The event fostered engaging discussions on common challenges in effective cancer care and therapeutic advancements. It also highlighted specific issues in oncology care in Latin America and showcased transferable strategies to address these challenges. Moreover, according to authors, the Summit successfully established a network of oncologists, hematologists, and scientists across Latin America, identifying opportunities for collaboration to improve cancer care regionally and globally through drug development and clinical research. This report summarizes key discussions from the Summit for the global and Latin American oncology community.
Comments:
Overview and general recommendation:
Check for plagiarism for similarity index. Yet, proofreading can enhance the quality of the manuscript. Several sentences need rewriting to make the readers comfortable when reading this. Avoid spelling errors.
1. There is a published report pertinent to this concept on the development of novel therapies and effective clinical patient-directed outcomes of cancers. however, Can you justify how expansion of physicians-directed policies beneficial in this oncology-based therapeutic regimens?
2. Authors should expand the discussion part with additional content.
3. Good to see authors have given ‘substantial information on combination therapies’ as well as emerging CART-therapies.
4. I am satisfied with other contents. Authors can enhance the script additional references and Figures to support their review hypothesis.
5. I am suggesting authors to include the radiation-induced Leukemia cancers and their discussions in their summit pertinent to mitigation strategies. For example, if a sudden nuclear attack takes place, the bone marrow will be affected with AML leukemia and in case of children, several pediatric cancers may occur due to lethal acute radiation exposure, how can you mitigate them ?? have you discussed these policies by the governments in your summit. This is crucial part of research needed to military and public benefit.
6. Conclusion should be explained vividly.
**Thank you**
Comments on the Quality of English Language
need to be improved
Round 2
Reviewer 2 Report
Comments and Suggestions for Authors
Authors addressed comments suggested by me
Comments on the Quality of English LanguagePerform at least 3 to 4 times proofreading to avoid any conceptual errors